# Applying Explainable Machine Learning Models for Detection of Breast Cancer Lymph Node Metastasis in Patients Eligible for Neoadjuvant Treatment

**DOI:** 10.3390/cancers15030634

**Published:** 2023-01-19

**Authors:** Josip Vrdoljak, Zvonimir Boban, Domjan Barić, Darko Šegvić, Marko Kumrić, Manuela Avirović, Melita Perić Balja, Marija Milković Periša, Čedna Tomasović, Snježana Tomić, Eduard Vrdoljak, Joško Božić

**Affiliations:** 1Department of Pathophysiology, University of Split School of Medicine, 21000 Split, Croatia; 2Department of Biophysics, University of Split School of Medicine, 21000 Split, Croatia; 3Department of Physics, University of Zagreb Faculty of Science, 10000 Zagreb, Croatia; 4Sigmoid Lab, Postindustria Group, 21000 Split, Croatia; 5Department of Pathology, University Hospital of Rijeka, 51000 Rijeka, Croatia; 6Department of Pathology, Clinical Hospital Sestre Milosrdnice, 10000 Zagreb, Croatia; 7Department of Pathology, Clinical Hospital Zagreb, 10000 Zagreb, Croatia; 8Department of Pathology, Clinical Hospital Dubrava, 10000 Zagreb, Croatia; 9Department of Pathology, University Hospital of Split, 21000 Split, Croatia; 10Department of Oncology, University Hospital of Split, 21000 Split, Croatia

**Keywords:** machine learning, breast cancer, neoadjuvant systemic treatment, lymph node metastasis

## Abstract

**Simple Summary:**

In this study, we trained and evaluated several machine-learning models with the aim of predicting breast cancer lymph node metastasis in patients eligible for neoadjuvant treatment. In neoadjuvantly treated patients, radiological and clinical methods are primary ways for determining axillary lymph node status, and radiological methods misdiagnose up to 30% of the patients. Hence, there is an unmet need for supplementary methods to aid oncologists and their multidisciplinary teams in assessing metastatic lymph node status and, consecutively, defining optimal treatment strategies. Good performance was achieved with a random forest algorithm (AUC: 0.79). We explored model explainability and, through it, exhibited how the models learned genuine relationships that were determined in previous studies. Such models can lead to more accurate disease stage prediction and consecutively better treatment selection, especially for NST patients, where radiological and clinical findings are often the only way of lymph node assessment.

**Abstract:**

Background: Due to recent changes in breast cancer treatment strategy, significantly more patients are treated with neoadjuvant systemic therapy (NST). Radiological methods do not precisely determine axillary lymph node status, with up to 30% of patients being misdiagnosed. Hence, supplementary methods for lymph node status assessment are needed. This study aimed to apply and evaluate machine learning models on clinicopathological data, with a focus on patients meeting NST criteria, for lymph node metastasis prediction. Methods: From the total breast cancer patient data (*n* = 8381), 719 patients were identified as eligible for NST. Machine learning models were applied for the NST-criteria group and the total study population. Model explainability was obtained by calculating Shapley values. Results: In the NST-criteria group, random forest achieved the highest performance (AUC: 0.793 [0.713, 0.865]), while in the total study population, XGBoost performed the best (AUC: 0.762 [0.726, 0.795]). Shapley values identified tumor size, Ki-67, and patient age as the most important predictors. Conclusion: Tree-based models achieve a good performance in assessing lymph node status. Such models can lead to more accurate disease stage prediction and consecutively better treatment selection, especially for NST patients where radiological and clinical findings are often the only way of lymph node assessment.

## 1. Introduction

Breast cancer is the most common cancer in women and contributes the most to women’s cancer mortality, which defines its public health importance [1,2]. Notably, 95% of newly diagnosed breast cancer patients are diagnosed in an early, locoregional, non-metastatic stage of the disease, when a cure is a realistic goal [3].

After the introduction of adjuvant and neoadjuvant therapies in the treatment of early breast cancer, the probability of 5-year survival almost doubled in the last 50 years [4]. Prognostic and predictive factors, biomarkers, and, recently, genetic panels are crucial in defining an optimal treatment strategy [5]. Still, one of the most important prognostic and treatment decision-making factors is the positivity of metastatic axillary lymph nodes [6].

Due to a recent change in the treatment strategy, significantly more patients are treated with neoadjuvant systemic therapy (NST). The NST is assigned based on the tumor biology characteristics and radiological findings [7]. The neoadjuvant concept allows in vivo testing of treatment sensitivity, further personalization of the adjuvant part of systemic therapy, and provides the way to receive accelerated approval of new treatments. It is also valid for the development of predictive biomarkers and reduces patient numbers in clinical trials by the usage of new surrogate endpoints such as the pathological complete response (pCR) [8,9].

It is known that axillary lymph node status is not precisely determined by radiological tests, ending up with close to 30% of patients being misdiagnosed and potentially not directed to NST [10].

After NST, the initial status of axillary lymph nodes is often questionable due to metastases completely responding to the NST, which consequently highlights the problem in defining treatment intensity for HER2 positive patients, for example, to continue or not with dual antiHER2 blockade with pertuzumab and trastuzumab [11].

Despite all the breakthroughs in our understanding of breast tumor biology, adjuvant/neoadjuvant therapy in early breast cancer therapy is still significantly based on the stage of the disease, whereas lymph node positivity usually prevents de-escalation of systemic therapy or omittance of lymph node radiotherapy [7]. Therefore, our knowledge about the tumor’s ability to metastasize and the actual status of lymph nodes is of paramount importance in our decision-making process.

Since NST is used more often and based on the rather limited accuracy of imaging methods applied in lymph node metastasis assessment, there is an unmet need for supplementary methods to aid oncologists and their multidisciplinary teams in the assessment of axillary lymph node status and consecutive definitions of optimal treatment strategies.

Due to exponential growth in oncology patient data, “Data Science” and machine learning techniques are extensively researched and applied as possible solutions to various clinical problems [12,13,14].

Machine learning, as a computational method that maps a mathematical function to a dataset in order to predict/classify the target variable, differs from traditional programming in that it directly learns from the data, without the need for explicit step-wise programming [15]. Traditional machine learning algorithms, such as support vector machines (SVM) and random forests (RF), have been successfully used to classify breast cancer into triple negative and non-triple negative types, predict the metastatic status of patients, and aid in detecting early disease recurrence [16,17]. Furthermore, more complicated models, such as the gradient-boosted trees and eXtreme Gradient Boosting (XGBoost), were used to predict survival outcomes in patients with epithelial ovarian cancer and the prediction of metastatic status in breast cancer, respectively [18,19]. The latest machine learning studies that focus on breast cancer achieved excellent performances, and are using deep learning techniques with radionics to classify breast cancer in radiological images or histopathological slides [20,21,22,23]. On the other hand, there are not many studies that utilize only clinicopathological features.

In this research, we trained and evaluated several machine learning models trained on multiple clinicopathological features obtained from the national-level breast cancer registry, with the goal of predicting patient axillary lymph node status accurately. By using a novel model explainability framework (SHAP), we presented how the model obtained its decision-making process. The study aims to evaluate explainable machine learning models for patients eligible for NST, as well as to assess how well the models can classify metastatic lymph nodes using only clinicopathological features.

This study’s main contributions are:machine learning model training, optimization, and evaluation curated specifically for patients eligible for NST;exhibiting what the model learned and which predictors were the most important in its decision-making process through the use of Shapley values;presenting model results for our whole breast cancer population (*n* = 8381).

## 2. Materials and Methods

### 2.1. Data Source and Preparation

Data examined in this study were collected from all Croatian hospitals in which breast cancer patients are diagnosed and treated. The data were acquired by searching through the hospital information systems during a five-year period, from January 2017 to January 2022. Pathohistological and demographic data were obtained for all patients that contained MKB code 50 (code for breast cancer). Pathohistological data was in a standardized format that follows ASCO/CAP guidelines, which all Croatian hospitals use [24].

The Ethics Committee of the University Hospital of Split approved the study protocol (2181-147/01/06/M.S.-22-02). The study was performed in accordance with the World Health Organization Declaration of Helsinki of 1975 as revised in 2013, and the International Conference on Harmonization Guidelines on Good Clinical Practice [25,26]. We fully protected the patients’ anonymity. The study was not preregistered.

The collected data consists of ten features: patient age at the time of diagnosis, tumor size (in cm), pathohistological type, immunophenotype, pathohistological grade, estrogen (ER) and progesterone (PR) receptor quantities (0–100), HER-2 levels (0–3), Ki-67 index (0–100), and lymph node metastasis status (0/1).

The case group was defined as patients with evidence of breast cancer axillary lymph node metastasis. Consequently, the control group was defined as patients without evidence of lymph node metastasis. Tumor samples were obtained via surgery and core needle biopsies, while the target variable ground truth (axillary lymph node positivity) was established by post-surgical lymph node pathohistological examination. While the tumor size was mostly obtained post-surgically, we argue how the model can also use radiologically determined tumor size (ultrasound, MRI, mammography, or CT), due to high concordance between the diagnostic methods and the final pathological measurement of tumor size (differences in tumor diameters <5 mm) [27,28].

Initial data set contained 13,580 entries, from which 3875 entries had various missing values, ranging from the target variable (lymph node metastasis status) to pathohistological type and grades. After we omitted the missing values, we were left with 9705 entries with complete data. From those 9705 entries, 1324 patients received neoadjuvant therapy, while 8381 received initial surgical treatment. We excluded 1324 neoadjuvantly treated patients from the analysis due to confounding effects of NST (NST would lead to lymph node negativity in up to 50% initially positive patients) [29].

Since the model’s target population is patients who would potentially receive neoadjuvant therapy, we identified those patients from our study population (all patients that initially received surgical treatment) using the following criteria:all tumors with size >5 cm (irrespective to subtype),tumors with size ≥2 cm of triple-negative or HER-2 positive subtype,tumors of inflammatory subtype [30].

By applying the NST criteria stated above, 719 patients were identified for final analysis.

In addition to the model based only on patients who would potentially receive neoadjuvant treatment, we also trained a broader model that generalizes to our entire breast cancer population (*n* = 8381), to see if similar performances are obtained and to analyze feature importance. Study workflow with methods for model optimization and validation is presented in Figure 1.

### 2.2. Prediction Model Training, Optimization and Validation

We trained different models using three algorithms: logistic regression, random forest classifier, and eXtreme gradient boosting (XGBoost) classifier. Random forest and XGBoost were selected because of their high-ranking performances on tabular data, whereas logistic regression was chosen as a base classifier for comparison [31,32]. Furthermore, for evaluation purposes, univariate logistic regression was trained only on one feature (tumor size) as a baseline.

For all models, we first split the data into a training (80% of data) and test batch (20% of data). The NST-criteria dataset is fairly balanced when concerning the target variable (55% vs. 45%), whereas due to the unbalance in total study population (34% vs. 66%) the train-test split was stratified on the target variable. To further compensate for the unbalanced target variable, we used threshold shifting (by maximizing the f1-score), and balanced class weights were used for the random forest (where weights are automatically adjusted inversely proportional to class frequencies in the input data).

Categorical variables were “dummy encoded” (encoding the categorical variables to ones and zeroes). We then performed a stratified 5-fold cross-validation on the training sample to train and validate our model (Figure 1). The model’s hyperparameters were optimized by performing a grid search (Figure 1). For the random forest we optimized the following hyperparameters: (1) maximal tree depth, (2) minimal number of samples required to split an internal node, (3) minimal number of samples required at a leaf node, (4) number of estimators. Whereas for XGBoost, the following hyperparameters were optimized: (1) maximal tree depth, (2) learning rate, (3) number of estimators, (4) minimum weight required to create a new node (“min_child_weight”), (5) gamma (pseudo-regularization parameter), (6) alpha (L1-regularization of leaf weights), (7) subsample (randomly selected training data prior to fitting to base learner), (8) subsample ratio of columns when constructing each tree (“colsample_bytree”), (9) subsample ratio of columns for each tree depth level (“colsample_bylevel”). Lastly, for logistic regression, we optimized for (1) solver (algorithm to use in the optimization problem), (2) regularization, and (3) regularization strength (C).

Additionally, since XGBoost can algorithmically deal with missing values, a subanalysis was performed on a dataset with missing values (after dropping the rows that miss the target variable; total *n* = 10,540, NST-criteria *n* = 1389).

Finally, the performance of the models was assessed on the test set, and the confidence intervals of the performance metrics were estimated using the bootstrap method of resampling with replacement (2000 bootstrapped samples). Modeling was performed using Python Programming Language (version 3.9.5, Python Software Foundation, Wilmington, DE, USA) using libraries “numpy”, “pandas”, “scikit-learn”, and “xgboost”, and with the R programming language (R Core Team, 2022, Vienna, Austria) using the “tidymodels”, “ranger”, “xgboost”, “pROC”, and “fastshap” packages.

### 2.3. Model Evaluation

Final evaluation and predictions were made on the test sample (20% of data). ROC curve was plotted, and areas under the curve (AUC) were obtained for each model with the following formula: $\mathrm{AUC}=\int_{0}^{1}\mathrm{TPR}\;\mathrm{dFPR}$, where $\mathrm{TPR}=\frac{\mathrm{TP}}{\mathrm{TP}+\mathrm{FN}}=\mathrm{sensitivity},\;$and $\mathrm{FPR}=\frac{\mathrm{FP}}{\mathrm{TN}+\mathrm{FP}}=1-\mathrm{specificity}$.

2000 bootstrap samples obtained by resampling with replacement from the test set were used to determine the mean AUC values and calculate the 95% confidence intervals. F1-score (harmonic mean between sensitivity and positive predictive value), precision (positive predictive value), negative predictive value, sensitivity and specificity were also determined. The model with the highest AUC was selected for further investigation. The optimal cut-off points for sensitivity and specificity were based on the F1 score [33].

### 2.4. Feature Importance Analysis and Model Explainability

We assessed feature importance by using SHAP (SHapley Additive exPlanations), a unified framework for interpreting model predictions [34]. The method computes Shapley values from coalitional game theory. The baseline for these values is the mean of all predictions. Shapley values explain how much each of the features moves the estimate from the baseline in order to obtain the final probability. When conditioning on a selected feature (predictor), the Shapely values attribute the change in the expected model prediction to that feature [34]. Hence, Shapley values can be used to explain machine learning model predictions.

### 2.5. Statistical Analysis

Descriptive statistical analysis was performed to analyze the characteristics of the case (positive lymph node) and control (negative lymph node) groups. Concerning numerical data, Student’s *t*-test was used to assess the comparison of means. Whereas for categorical variables, χ2 test was used. Calculations were performed with Python Programming Language (version 3.9.5, Python Software Foundation) using the “scipy” library. Statistical significance was set at *p* < 0.05 for all comparisons.

## 3. Results

### 3.1. Patient Characteristics

In total, 5845 (69.7%) patients were identified as controls (no lymph node metastasis), while 2536 (30.3%) patients were identified as cases (positive lymph node/s) (Table 1). Statistically significant differences between cases and controls were observed in “Tumor size”, “PR”, “HER-2”, and “Ki-67” features (Table 1).

In the NST criteria group, there were 426 (55%) patients with lymph node metastasis and 350 (45%) patients without metastasis. Since lymph node metastasis is present in 55% of the target population, we can see that our target variable is fairly balanced (55% vs. 45%), which differs from the total population where the ratio favors the non-metastasis group (34% vs. 66%).

Moreover, for the NST criteria group, while “Tumour size”, “PR” and “Ki-67” also showed significant differences, there were also significant differences in “Age” and “ER” features and no significant difference in “HER-2” (Table 2).

### 3.2. Prediction Model Performance

#### 3.2.1. Performance on NST Criteria Group

After training three different models on the NST-criteria group data (*n* = 621), validating and optimizing them via 5-fold cross-validation, and then evaluating them on the holdout test set (*n* = 155), the random forest classifier produced the highest result. (Table 3). Using default settings, the random forest classifier achieved an AUC of 0.76, whereas, after hyperparameter optimization, the score rose to 0.793 (95% CI 0.713–0.865) (Figure 2, Table 3). At the baseline decision threshold of 0.5, F1-score was 0.750 (95% CI: 0.690–0.812), sensitivity was 0.809 (95% CI 0.718–0.885), specificity 0.570 (95% CI 0.446–0.692), negative predictive value 0.714 (95% CI 0.615–0.820) and the precision (positive predictive value) 0.694 (95% CI: 0.630–0.759). Another tree-based model, XGBoost, achieved an AUC of 0.783 (95% CI: 0.703–0.858) on the test set (Table 3). Finally, Logistic Regression has achieved an AUC of 0.763 (95% CI: 0.683–0.838), while univariate Logistic Regression (trained on “Tumor size”) achieved an AUC of 0.688 (95% CI: 0.626–0.745) (Table 3).

When evaluating XGBoost on the NST-criteria dataset that contained missing values (*n* = 1389), somewhat worse performances were obtained, with an AUC of 0.724 (95% CI: 0.654–0.785).

#### 3.2.2. Performance on Entire Population

Due to a higher *n*, instead of 20%, we held out 10% of the data for the test set in the entire population. Therefore, 7543 rows of data were used for the training set, while 838 rows were held out in the test set. A 10-fold cross-validation scheme was performed on the training data to train and validate the models. Finally, their individual performances were assessed on the test set and standard deviations were obtained with the bootstrap method (Table 4). XGBoost ranked highest with a mean AUC of 0.762 (95% CI: 0.726–0.794), closely trailed by Random Forest with an AUC of 0.760 (95% CI: 0.71–0.78) (Table 4). Just like with Random Forest and XGBoost, Logistic Regression and Univariate Logistic Regression also scored lower than in the NST criteria group, with a mean AUC of 0.741 (95% CI: 0.706–0.775) and 0.589 (95% CI 0.577–0.614), respectively (Table 4). Concerning XGBoost’s performance on other metrics at the baseline threshold, it achieved an F1 score of 0.448 (95% CI: 0.389–0.507), a sensitivity of 0.344 (95% CI: 0.289–0.403) and specificity of 0.903 (95%: 0.877–0.926), the positive predictive value of 0.607 (95% CI: 0.539–0.680), the negative predictive value of 0.761 (95% CI: 0.744–0.778) (Figure 3). To correct the class imbalance we changed the default threshold by maximizing the F1 score. Lowering the threshold to 0.28 increased the F1-score to 0.581 (95% CI: 0.545–0.618), sensitivity to 0.732 (95% CI: 0.676–0.787), and negative predictive value to 0.854 (95% CI: 0.827–0.881), while specificity decreased to 0.676 (95% CI: 0.637–0.714), and positive predictive value to 0.495 (95% CI: 0.461–0.531). When evaluating XGBoost on a total dataset that contained missing values (*n* = 10,540), somewhat worse performances were obtained, with an AUC of 0.731 (95% CI: 0.634–0.771).

#### 3.2.3. Feature Importance for Predicting Lymph Node Metastasis

After calculating Shapely values for the NST criteria group, tumor size was the most important feature, followed by ER, PR, and HER2 status (Figure 4).

Lymph node metastasis showed a linear dependence on tumor size up to 5 cm, after which a plateau is reached (Figure 5). Concerning age, there is a clear and sharp rise in dependency after the age of 75 (Figure 5). ER and PR status show a growing trend, with larger values more associated with nodal involvement, while hormone receptor negativity is associated with an absence of metastasis (Figure 5). Notably, this is highly correlated to tumor size, because most high ER and PR tumors were of luminal A and luminal B histological types, which have to be >5 cm in size to adhere to NST criteria.

For HER-2, the model associated HER-2 positivity with a higher chance of lymph node metastasis (Figure 5). Interestingly, Ki-67 exhibits an increase in Shapley values from 0 to 25%, after which it gradually decreases, with a sudden drop in values at around 75%. However, after a more detailed inspection, we can see that these values are predominantly associated with the triple-negative immunophenotype (Figure 5). Finally, the model associated lobular invasive histological type with a higher chance of lymph node metastasis (Figure 5).

While tumor size was also the most important feature when Shapley values were calculated on the total study population, the second most important feature was Ki-67, followed by age and tumor grade (Figure 6).

For tumor size, the trend from the NST criteria group was confirmed on the total study population (growing dependency, with a plateau after 5 cm) (Figure 7). The same holds for HER-2 status, where HER-2 positivity is associated with lymph node metastasis (Figure 7). Tumor grade also shows a clear increasing trend (Figure 7). For Ki-67, there is a noticeable increase in Shapley values at index values of 25%, along with a decrease at around 75%. However, the Shapley values exhibit a high level of dispersion (Figure 7), indicating a dependence on the value of other variables. Age shows an interesting non-linear dependency, where patients younger than 40 and patients older than 75 were associated with a higher chance of metastasis (Figure 7).

## 4. Discussion

In order to improve treatment outcomes, an increasing number of early breast cancer patients are treated with NST. To achieve optimal treatment strategy for every patient, and implement precision oncology, due to nonoptimal performance of existing diagnostic tools, there is a growing need for additional methods of axillary lymph node metastasis status evaluation. Additionally, to determine axillary lymph node status in patients with breast cancer who are initially treated with surgical therapy, sentinel lymph node dissection is recommended. It was reported that sentinel lymph node dissection has a usual false negative range of 7.5%, but can reach up to 27.3%, resulting in unradical axillary surgery and consequently suboptimal adjuvant therapy strategies based on wrong staging [35,36]. Therefore, other than patients receiving NST, patients receiving initial surgical therapy can also benefit from supplementary non-invasive methods for the determination of the lymph node status.

In this study, we trained, optimized, and validated multiple machine-learning models that can effectively help us predict axillary breast cancer lymph node metastasis. The random forest algorithm, an ensemble-based method, has produced the highest mean AUC score (0.793 for the NST criteria group, 0.760 for the total study population). Since the random forest algorithm consists of multiple decision trees and later combines their predictions, it reduces the risk of overfitting and thus provides a more robust model [37]. XGBoost, another robust ensemble tree-based model, produced a somewhat better result than random forest on the total study population. This is in line with previous studies showing that XGBoost and Random forest generally achieve comparable results, but based on the exact dataset, one will outperform the other [16,38]. Lastly, both models achieved improved performances when compared to the baseline univariate model that only used tumor size as a predictor.

When considering feature importance by Shapley values, the most important features in the NST criteria group were tumor size, ER, PR and HER2 status. Nodal involvement grew with tumor size up to 5 cm, and hence our findings corroborate the findings of Sopik and Narod, where a plateau in nodal metastasis was also reached after approximately 5 cm in tumor size [39]. For the NST-criteria group, Shapley values display a growing trend with an increase in ER and PR levels. Although this is surprising at first glance, this finding is probably an artifact of selection criteria. Namely, based on the selection criteria, tumors with high ER and/or PR values, corresponding to luminal A and B immunophenotypes, were only included if the tumor size was greater than 5 cm. This leads to an artificial correlation between high ER values and the probability of lymph node metastasis, which is actually based on the influence of tumor size. Consequently, the results concerning the effect of ER and PR were not confirmed when feature importance was examined in the total study population. The literature is also conflicted concerning the role of ER in predicting nodal involvement. For example, in a study by Alsumai et al., positive ER status was a significant predictor of nodal metastasis, whereas in another study ER and PR held no significant effect on axillary lymph node metastasis [40,41].

Furthermore, Shapley values for patient age showed that nodal involvement grows after 75 years of age. Similar results were reported by Wildiers et al., where nodal involvement grew after the age of 70 [42].

While tumor size was also the most important feature in the total study population, it was shown that the second most important feature was Ki-67, followed by tumor grade. Interestingly, when Shapley values for patient age were calculated on the total study population, a non-linear trend was observed, where both patients younger than 40 and patients older than 75 were associated with nodal involvement. This higher occurrence of lymph node metastasis in patients younger than 40 was also reported in previous studies [43,44]. Concerning Ki-67, higher values are generally associated with higher probabilities for metastasis. Interestingly, for Ki-67 values > 75%, Shapley values are mostly negative. However, when accounting for immunophenotype, this trend reversal is only visible for triple-negative carcinomas. Previous studies also showed that Ki-67 > 20% was positively correlated with lymph node metastasis, albeit they used pre-defined categories of <20% and >20% and did not account for differences in immunophenotype [45,46]. Hence, they potentially missed a more complex relationship that was explained through Shapley values.

Likewise, higher tumor grade is commonly associated with nodal metastasis [43]. Concerning HER-2, our model associates HER-2 positivity with nodal involvement. HER-2 was also a significant predictive factor for axillary nodal involvement (with a regression coefficient of 0.30) in another study where researchers developed a Lasso regression model to predict non-sentinel breast cancer lymph node metastasis [47]. Taken together, we can conclude that our model’s decision-making process can be clinically explained because it has learned relationships whose importance was also confirmed by other studies.

Aside from the above observations, it is also worth noting that Shapley analysis has independently identified the commonly used cutoff points for ER, PR, Ki-67, and HER2 positivity.

Currently, we are witnessing a great interest in studies based on radiomics, combining radiological findings and deep learning methods to predict breast cancer lymph node status [48,49,50,51]. An interesting study by Zheng et al. joined deep learning radiomics of conventional ultrasound and shear wave elastography of breast cancer with clinicopathological data and obtained excellent results with an AUC of 0.902 (95% CI: 0.843–0.961) [52]. They also applied a model that was trained just with features from clinicopathological data, which produced a weaker AUC in comparison to ours (0.72 [95% CI: 0.63–0.82] vs. 0.79 [95% CI 0.72–0.87], respectively) [52]. This relatively poorer artificial neural networks (ANN) performance could potentially be explained by a relative underperformance of neural networks on tabular data [31]. It was shown that tree-based models (random forest and XGBoost) outperform deep learning methods on datasets with up to 10 000 training examples [31].

Similarly, another study that evaluated ANN trained on clinicopathological features for predicting breast cancer lymph node involvement achieved an AUC of 0.74 (95% CI: 0.72–0.76) [53]. However, one of their most predictive features was a lymphovascular invasion, a feature that is not always obtainable on core-needle biopsy and is not used by our model [54].

Random forest was also the best-performing algorithm in the study by Tseng et al., with performances similar to ours (mean AUC of 0.75) [17]. Another tree-based model (XGBoost), was shown to be the best-performing model in a study by Li et al., where the authors utilized tumor gene signatures to predict metastatic status in breast cancer [19]. Their optimized model achieved an AUC of 0.82 (SD ± 0.15) [19]. Likewise, in a study by Meng et al., a Lasso regression-based model achieved an AUC of 0.77 (95% CI 0.69–0.86) for the prediction of non-sentinel lymph node metastasis status [47]. Moreover, an older study that combined clinicopathological findings with diagnostic mammography and ultrasonography findings achieved an AUC of 0.77 (95% CI: 0.689–0.856) in breast cancer lymph node prediction with an alternating decision tree (ADTree) [55]. Therefore, our results are comparable with the results of other studies that predicted breast cancer lymph nodes and general metastasis [17,19,47,52,55] (Table 5).

We believe that similar models could be locally optimized and validated to aid clinicians in their multidisciplinary workflow. Especially when dealing with patients who would receive NST, since lymph node status is an essential factor that affects optimal treatment selection and prognosis. Moreover, other beneficial tumor/patient data that can be obtained, such as gene expression and serum biomarkers, could lead to better model performances. Accordingly, future research can assess whether the addition of genetic and biomarker data increases the accuracy of machine-learning models.

Our study contains several limitations. Firstly, it was performed only on the Croatian population of early breast cancer patients. Thus, similar models should be validated on other population groups to provide better generalizability. Another limitation of this study is its retrospective nature, even though the data originated from a prospectively maintained database. Perhaps the most important limitation of the study was the relatively large number of patients who were excluded from the analysis due to incomplete data. Of course, we have no evidence that these data are missing completely at random (MCAR). Furthermore, a possible minor limitation of the generalizability of the results of this analysis could have been caused by the fact that part of the data was collected during the lockdown to control the COVID-19 pandemic.

Finally, this study offers novelty by presenting an explainable machine-learning framework with a clinically relevant decision-making process. A further strength of the present study is that it provides a unique perspective in which a multicenter dataset was obtained, and from subjects that were initially treated surgically, an additional subset was extracted by applying the NCCN criteria for NST [7]. In this way, the model was curated for NST-eligible patients, who could extract the greatest benefit from such a non-invasive method for determining axillary lymph node metastasis status.

## 5. Conclusions

We have shown that explainable tree-based machine learning methods trained on patient and tumor features obtained during regular pre-operative/pre-NST procedures achieve a good performance in predicting breast cancer axillary lymph node metastasis. Such models can lead to more accurate diagnosis and better treatment selection, especially for NST patients, where radiological and clinical findings are often the only way of lymph node assessment. Potential upstage of diagnosis based on machine learning models for some patients would result in NST and, consecutively, potentially more adjuvant therapy with non-cross resistant treatments and better patient outcomes. The addition of genetic and biomarker data and subsequent validation in multinational/multicenter studies is expected from future studies.

## Figures and Tables

**Figure 1 cancers-15-00634-f001:**
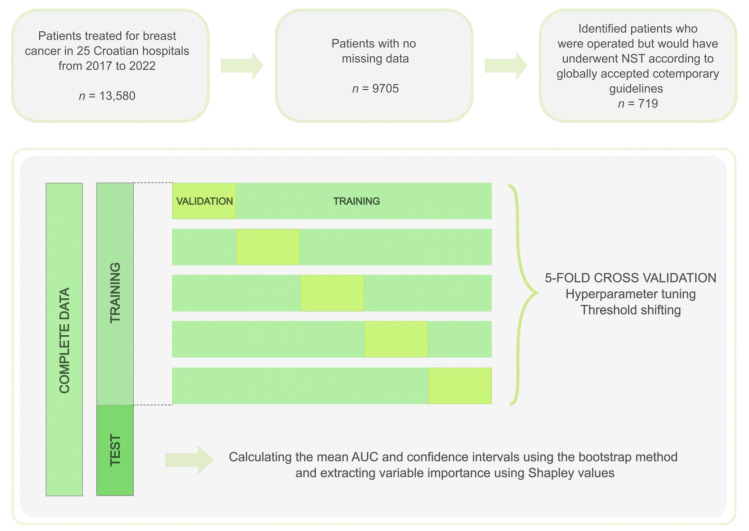
Study workflow.

**Figure 2 cancers-15-00634-f002:**
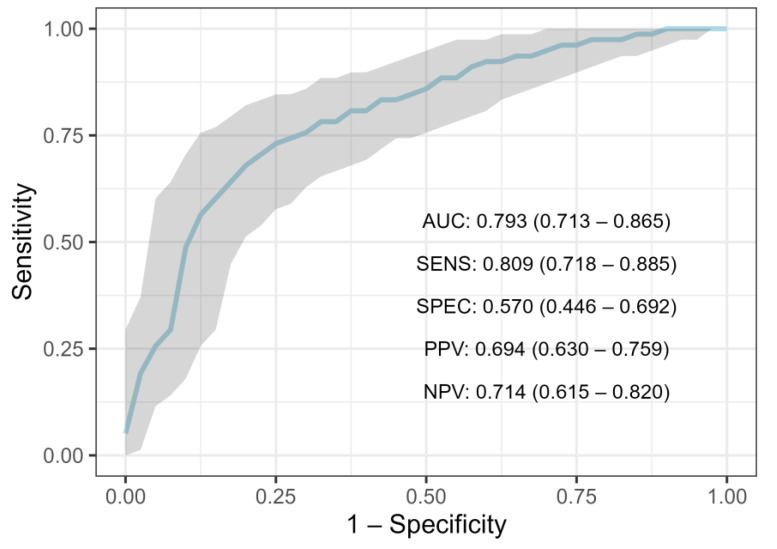
ROC-curve for Random Forest trained on NST-criteria group. (ROC—receiver operating characteristic curve, AUC—area under the curve, SENS—sensitivity, SPEC—specificity, PPV—positive predictive value, NPV—negative predictive value).

**Figure 3 cancers-15-00634-f003:**
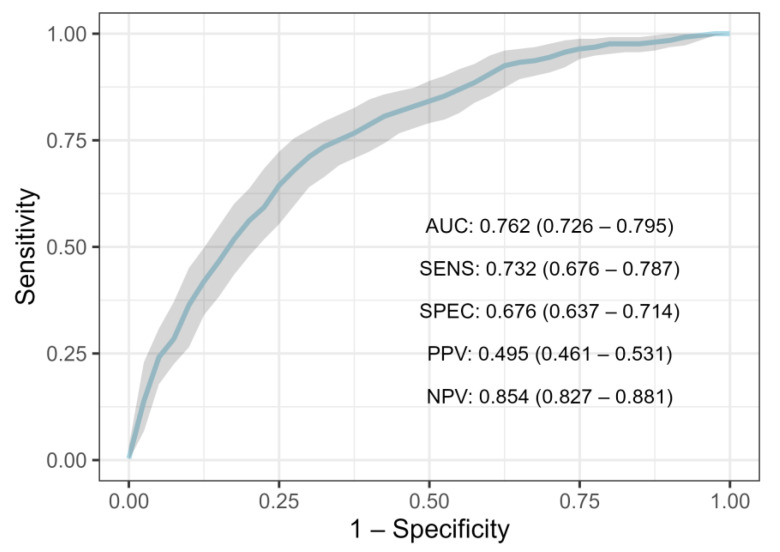
ROC-curve for XGBoost trained on total study population. (ROC—receiver operating characteristic curve, AUC—area under the curve, SENS—sensitivity, SPEC—specificity, PPV—positive predictive value, NPV—negative predictive value).

**Figure 4 cancers-15-00634-f004:**
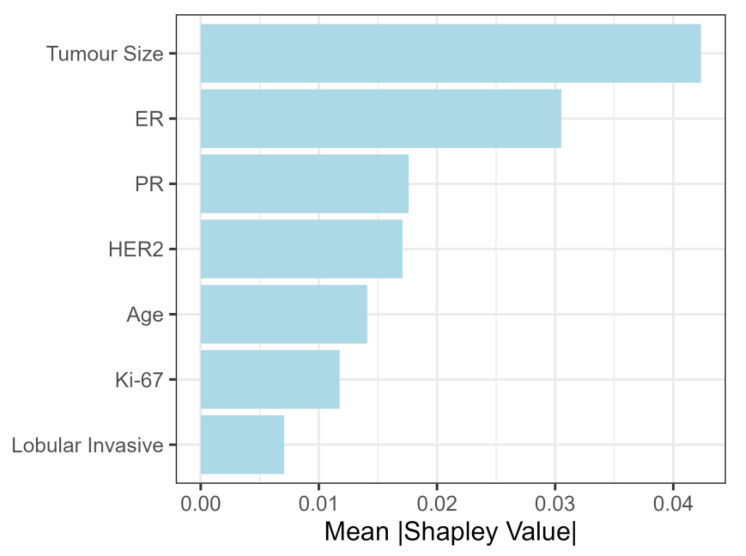
Shapley values (feature importance) for NST group.

**Figure 5 cancers-15-00634-f005:**
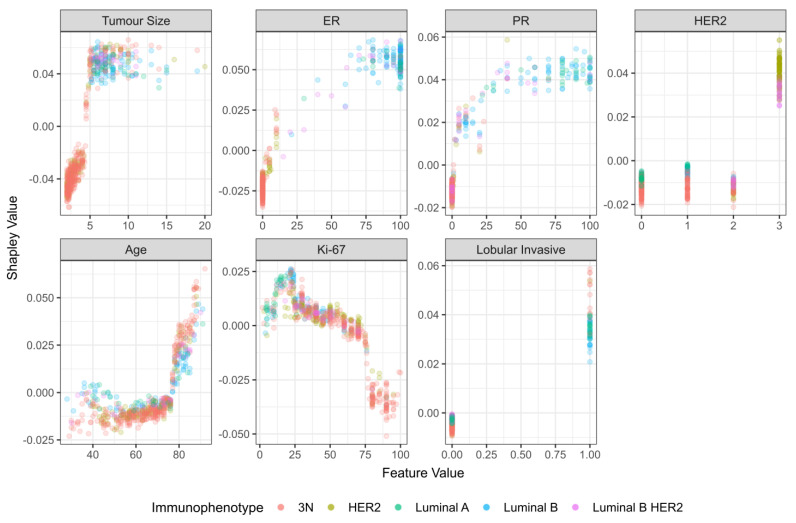
Dependency plot for Shapley values (NST criteria group).

**Figure 6 cancers-15-00634-f006:**
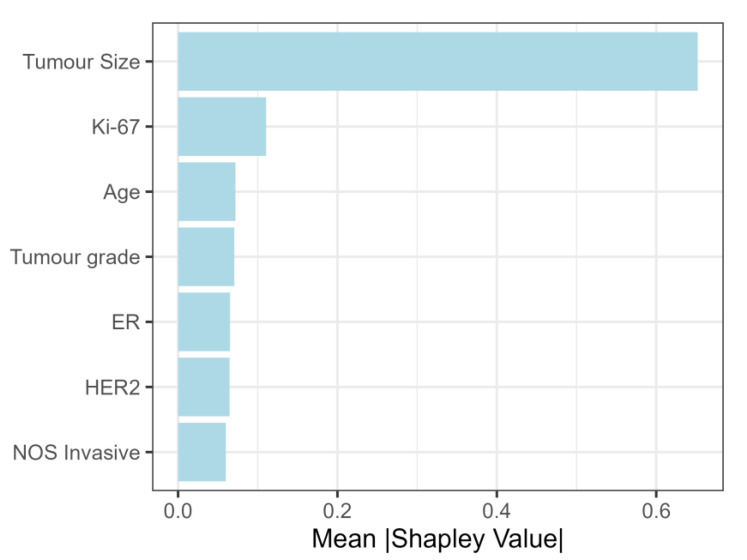
Shapley values (feature importance) for total study population.

**Figure 7 cancers-15-00634-f007:**
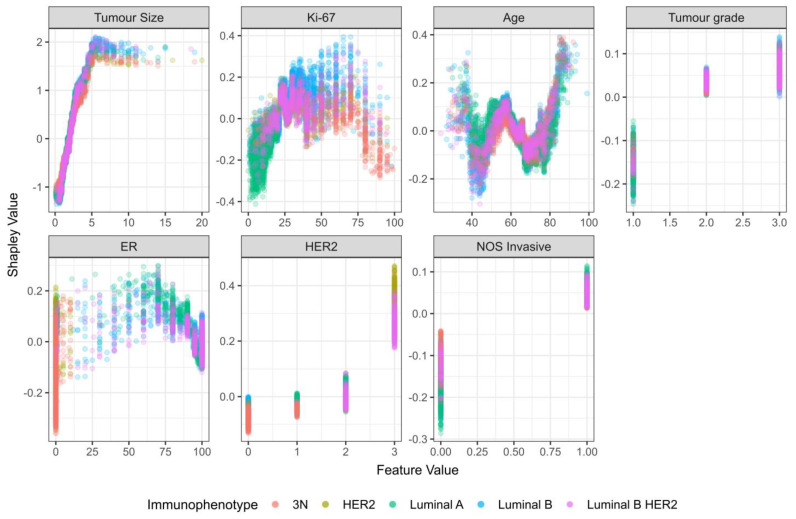
Dependency plot for Shapley values (total study population).

**Table 1 cancers-15-00634-t001:** Patient characteristics (total population case/control comparison).

Variable	Cases, Lymph Node Metastasis Group (*n* = 2536)	Controls, Non-Lymph Node Metastasis Group (*n* = 5845)	Total (*n* = 8381)	*p*-Value
Age (range)	63.6 (21–92)	62.3 (25–89)	62.7 (12.6)	0.535 *
Tumor Size (cm)	2.7 (1.9)	1.7 (1.1)	2.01 (1.5)	<0.001 *
Ki-67	29.7 (18.7)	25.1 (18.4)	26.5 (18.7)	<0.001 *
ER	80.7 (33.7)	83.1 (32.2)	82.4 (32.7)	0.064 *
PR	50.8 (39.5)	54.3 (39.4)	53.2 (39.4)	<0.001 *
Tumor Grade (%)				<0.001 †
1	376 (14.8)	1638 (28)	2014 (24)	
2	1460 (57.6)	3148 (53.8)	4608 (54.9)	
3	700 (27.6)	1059 (18.2)	1759 (20.9)	
HER-2 (%)				<0.001 *
0	1092 (43.1)	2776 (47.5)	3868 (46.2)	
1	815 (32.1)	1953 (33.4)	2768 (33)	
2	344 (1.3)	629 (10.8)	914 (10.9)	
3	285 (11.2)	487 (8.3)	831 (9.9)	
Histological Type (%)				<0.001 †
NOS-invasive	2055 (81)	4586 (78.5)	6641 (79.2)	
Lobular Invasive	324 (12.8)	693 (11.9)	1017 (12.1)	
Ca with Medullary Characteristics	24 (0.9)	47 (0.8)	71 (0.8)	
Other (Rare Types)	133 (5.2)	519 (8.9)	652 (7.8)	
Immunophenotype (%)				<0.001 †
Luminal B	1517 (59.8)	3154 (53.9)	4671 (55.7)	
Luminal A	429 (16.9)	1628 (27.8)	2057 (24.6)	
Luminal B-her2	310 (12.3)	508 (8.8)	818 (9.8)	
Triple Negative	160 (6.3)	407 (6.9)	567 (6.7)	
HER2 Positive	120 (4.7)	148 (2.6)	268 (3.2)	

Data are presented as mean (standard deviation) and count (percentage); *—*t*-test for independent variables, †—χ^2^ test; Ki-67—cellular proliferation index, ER—estrogen receptor index, PR—progesterone receptor index, HER-2—human epidermal growth factor receptor, NOS- not otherwise specified histological type. Other rare histological types include: mucinous invasive, micropapillary invasive, cribriform invasive, and inflammatory types).

**Table 2 cancers-15-00634-t002:** Patient characteristics (NST criteria group).

Variable	Cases, Lymph Node Metastasis Group (*n* = 392)	Controls, Non-Lymph Node Metastasis Group (*n* = 327)	Total (*n* = 719)	*p*-Value
Age (range)	66.9 (21–92)	64.2 (25–87)	65.7 (14.6)	0.016 *
Tumor Size (cm)	5.7 (3.02)	3.9 (2.5)	4.9 (2.9)	<0.001 *
Ki-67	43.4 (23.3)	50.6 (25.7)	46.7 (24.7)	<0.001 *
ER	40.01 (45.9)	13.7 (32.6)	28.1 (42.5)	<0.001 *
PR	21.8 (35.04)	7.51 (22.5)	15.2 (30.6)	<0.001 *
Tumor Grade (%)				0.027 †
1	14 (3.6)	8 (2.4)	22 (3)	
2	133 (33.9)	97 (29.6)	230 (32)	
3	245 (62.5)	222 (68)	467 (65)	
HER-2 (%)				0.060 *
0	180 (46)	173 (53)	353 (49.1)	
1	79 (20.1)	55 (16.8)	134 (18.7)	
2	35 (8.9)	42 (12.8)	77 (10.7)	
3	98 (25)	57 (17.4)	155 (21.5)	
Histological Type (%)				<0.001 †
NOS-invasive	282 (71.9)	252 (77.2)	534 (74.3)	
Lobular Invasive	63 (16.1)	20 (6.1)	83 (11.5)	
Ca with Medullary Characteristics	15 (3.8)	12 (3.6)	27 (3.7)	
Other (Rare Types)	32 (8.2)	43 (13.1)	75 (10.4%)	
Immunophenotype (%)				<0.001 †
Luminal B	111 (28.4)	35 (10.7)	146 (20.3)	
Luminal A	26 (6.6)	7 (2.3)	33 (4.6)	
Luminal B-her2	35 (8.9)	8 (2.4)	43 (5.9)	
>Triple Negative	137 (34.9)	210 (64.2)	347 (48.4)	
HER-2 Positive	83 (21.2)	67 (20.4)	150 (20.8)	

Data are presented as mean (standard deviation) and count (percentage); *—*t*-test for independent variables, †—χ^2^ test;).

**Table 3 cancers-15-00634-t003:** Model performances for predicting lymph node metastasis (NST criteria group).

Model	Mean AUC (95% CI)
Random Forest	0.793 (0.713–0.865)
XGBoost	0.783 (0.703–0.858)
Logistic Regression	0.763 (0.683–0.838)
Univariate Logistic Regression	0.645 (0.556–0.726)

Values are presented as mean (95% Confidence interval); AUC—area under the receiver operating characteristic curve.

**Table 4 cancers-15-00634-t004:** Model performances for predicting lymph node metastasis (entire population).

Model	Mean AUC (95% CI)
XGBoost	0.762 (0.726–0.795)
Random Forest	0.760 (0.724–0.794)
Logistic Regression	0.741 (0.706–0.775)
Univariate Logistic Regression	0.713 (0.686–0.739)

Values are presented as mean (95% Confidence interval); AUC—area under the receiver operating characteristic curve.

**Table 5 cancers-15-00634-t005:** Comparison with other studies that used clinicopathological features for breast cancer lymph node classification.

Study (Algorithm Type)	Total Patients	Mean AUC (95% CI)
This study (XGBoost)	8381	0.76 (0.73–0.80)
Takada et al. [55] (ADTree)	467	0.77 (0.69–0.86)
Zheng et al. [52] (without radiomics, neural network)	1342	0.72 (0.63–0.82)
Dihge et al. [53] (neural network)	800	0.74 (0.72–0.76)
Meng et al. [47] (non-sentinel lymph node prediction, Lasso regression)	714	0.77 (0.69–0.86)

## Data Availability

The dataset used and analyzed during the current study is available from the corresponding author upon reasonable request.

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
