# Peer review of "Applying Explainable Machine Learning Models for Detection of Breast Cancer Lymph Node Metastasis in Patients Eligible for Neoadjuvant Treatment"

_cancers, 2023, doi:10.3390/cancers15030634_

Round 1

Reviewer 1 Report

The study is an interesting one. I have only a few suggestions.

1. Main contributions of the study may he highlighted point-wise in the introduction section.

2. Research questions and motivation of the study may be elaborated more.

3. Please justify the utilization of only tree-based classifiers whereas there are lots of other machine learning algorithms available. If tree-based classifiers had yielded good performance, then which were these studies?

4. Was the dataset balanced? If not, what was the method applied?

5. Limitations and future works of the study may be incorporated.

6. Comaprison of the study with the SOTA studies may be added in a tabular form.

7. Recent reviews and studies may be cited such as given below:

Iqbal, M. S., Ahmad, W., Alizadehsani, R., Hussain, S., & Rehman, R. (2022, November). Breast Cancer Dataset, Classification and Detection Using Deep Learning. In Healthcare (Vol. 10, No. 12, p. 2395). MDPI.  

Author Response

Dear reviewer, thank you for your comments. 

1) Main contributions of the study may he highlighted point-wise in the introduction section.

Thank you for the constructive suggestion with which we absolutely agree. Concerning the main contributions of the study, we added the following at the end of introduction:

“This study’s main contributions are 1) machine learning model training, optimization and evaluation curated specifically for patients eligible for NST; 2) exhibiting what the model learned and which predictors were the most important in its decision-making process through the use of Shapley values; 3) presenting model results on our whole breast cancer population (N= 8381)

(lines 114-119)

2) Research questions and motivation of the study may be elaborated more.

Thank you for your valued comment.

Concerning the research questions and study motivation, we added the following to the introduction: “The study aims to evaluate explainable machine learning models for patients eligible for NST, as well as to assess how well the models can classify metastatic lymph nodes using only clinicopathological features.” (lines 111-113)

3) Please justify the utilization of only tree-based classifiers whereas there are lots of other machine learning algorithms available. If tree-based classifiers had yielded good performance, then which were these studies?

Thank you for this observation.

Concerning the justification of using tree-based algorithms and their SOTA performance on tabular data, we have added the following references:

  1. “Grinsztajn, L.; Oyallon, E.; Varoquaux, G. Why do tree-based models still outperform deep learning on tabular data? arXiv preprint arXiv:2207.08815 2022.” (reference stated in line 171)
  2. Shwartz-Ziv, R.; Armon, A. Tabular data: Deep learning is not all you need. Information Fusion 2022, 81, 84-90, doi:https://doi.org/10.1016/j.inffus.2021.11.011.

  • Was the dataset balanced? If not, what was the method applied?

Thank you for your valued comment.

Concerning whether the dataset was balanced- In the results section (rows 205-209) we state the following “In the NST criteria group, there were 426 (55 %) patients with lymph node metastasis and 350 (45 %) patients without metastasis. Since lymph node metastasis is present in 55 % of target population, we can see that our target variable is fairly balanced (55 % vs 45 %), which differs from the total population where the ratio favours the non-metastasis group (34 % vs 66 %).” (lines 246-250)

Furthermore, we have added the following to the methods section:

“The NST-criteria dataset is fairly balanced when concerning the target variable (55 % vs 45 %), whereas due to the unbalance in total study population (34 % vs 66 %) the train-test split was stratified on the target variable. To further compensate for the unbalanced target variable we used threshold shifting (by maximizing the f1-score), and balanced class weights were used for the random forest (where weights are automatically adjusted inversely proportional to class frequencies in the input data).” (lines 175-180)

4) Limitations and future works of the study may be incorporated.

Thank you for your valued comment.

Concerning future works and limitations, the same are expressed in the discussion (lines 400-416):

“We believe that similar models could be locally optimized and validated to aid clinicians in their multidisciplinary workflow. Especially when dealing with patients who would receive NST, since lymph node status is an essential factor that affects optimal treatment selection and prognosis. Moreover, other beneficial tumour/patient data that can be obtained, such as gene expression and serum biomarkers, could lead to better model performances. Accordingly, future research can assess whether the addition of genetic and biomarker data increases the accuracy of machine learning models.

Our study contains several limitations. Firstly, it was performed only on the Croatian population of early breast cancer patients. Thus, similar models should be validated on other population groups to provide better generalizability. Another limitation of this study is its retrospective nature, even though the data originated from a prospectively maintained database. Perhaps, the most important limitation of the study was the relatively large number of patients who were excluded from the analysis due to incomplete data. Of course, we have no evidence that these data are missing completely at random (MCAR). Furthermore, a possible minor limitation of the generalizability of the results of this analysis could have been caused by the fact that part of the data was collected during the lockdown to control the COVID-19 pandemic”

5) Comparison of the study with the SOTA studies may be added in a tabular form.

Thank you for the helpful and constructive input.

Concerning the comparison of the study with the SOTA studies, we added Table 5. in the discussion. (lines 449-451)

6) Recent reviews and studies may be cited such as given below:

Iqbal, M. S., Ahmad, W., Alizadehsani, R., Hussain, S., & Rehman, R. (2022, November). Breast Cancer Dataset, Classification and Detection Using Deep Learning. In Healthcare (Vol. 10, No. 12, p. 2395). MDPI.

Thank you for your valued comment.

We have added the following reference (references stated in lines 90-92, and 104-106)

“14. Iqbal, M. S., Ahmad, W., Alizadehsani, R., Hussain, S., & Rehman, R. (2022, November). Breast Cancer Dataset, Classification and Detection Using Deep Learning. In Healthcare (Vol. 10, No. 12, p. 2395). MDPI.” in the introduction.

Furthermore, we added the following also in the introduction:

“20. Altameem, A.; Mahanty, C.; Poonia, R.C.; Saudagar, A.K.J.; Kumar, R. Breast Cancer Detection in Mammography Images Using Deep Convolutional Neural Networks and Fuzzy Ensemble Modeling Techniques. Diagnostics 2022, 12.”

“21. Muduli, D.; Dash, R.; Majhi, B. Automated diagnosis of breast cancer using multi-modal datasets: A deep convolution neural network based approach. Biomedical Signal Processing and Control 2022, 71, 102825, doi:https://doi.org/10.1016/j.bspc.2021.102825.”

“22. Wakili, M.A.; Shehu, H.A.; Sharif, M.H.; Sharif, M.H.U.; Umar, A.; Kusetogullari, H.; Ince, I.F.; Uyaver, S. Classification of Breast Cancer Histopathological Images Using DenseNet and Transfer Learning. Comput Intell Neurosci 2022, 10.”

“23. Heenaye-Mamode Khan, M.; Boodoo-Jahangeer, N.; Dullull, W.; Nathire, S.; Gao, X.; Sinha, G.R.; Nagwanshi, K.K. Multi- class classification of breast cancer abnormalities using Deep Convolutional Neural Network (CNN). PLoS One 2021, 16.”

Reviewer 2 Report

The authors developed and evaluated several machine-learning methods for breast cancer lymph node metastasis in patients. The results show that using the Random Forest method provides good results and performance. The paper should be improved before publication, and my major comments are as follows:

1) Novelty and Contribution are unclear and must be emphasized in the paper. As a suggestion, the authors may open a new subsection and describe the Novelty and Contribution of the paper.

2) In the second section, the authors should provide more information about the dataset.

3) The related work section should be improved by including the most recent papers related to breast cancer detection and classification. For instance,

MITNET: a novel dataset and a two-stage deep learning approach for mitosis recognition in whole slide images of breast cancer tissue

Breast Cancer Detection in Mammography Images Using Deep Convolutional Neural Networks and Fuzzy Ensemble Modeling Techniques

Automatic Detection and Classification of Mammograms Using Improved Extreme Learning Machine with Deep Learning

Classification of Breast Cancer Histopathological Images Using DenseNet and Transfer Learning

Fuzzy ensemble of deep learning models using choquet fuzzy integral, coalition game and information theory for breast cancer histology classification

Automated diagnosis of breast cancer using multi-modal datasets: A deep convolution neural network based approach

An automated deep learning based mitotic cell detection and recognition in whole slide invasive breast cancer tissue images

Multi- class classification of breast cancer abnormalities using Deep Convolutional Neural Network (CNN)

4) The authors should enhance the quality of the images in the results section. Also include the equation for AUC.

5) The authors included parameters description for machine learning methods, but I couldn't find a description of the parameters. The authors should include this information in the paper.

Author Response

Dear reviewer, Thank you for your dedicated time and helpful comments. 

1) Novelty and Contribution are unclear and must be emphasized in the paper. As a suggestion, the authors may open a new subsection and describe the Novelty and Contribution of the paper.

Thank you for your valued comment.

We added a new paragraph in the introduction that better describes the novelty and contributions of the paper (lines 114 – 119):

“The study aims to evaluate explainable machine learning models for patients eligible for NST, as well as to assess how well the models can classify metastatic lymph nodes using only clinicopathological features.

 This study’s main contributions are 1) machine learning model training, optimization and evaluation curated specifically for patients eligible for NST; 2) exhibiting what the model learned and which predictors were the most important in its decision-making process through the use of Shapley values; 3) presenting model results on our whole breast cancer population (N= 8381).”

2) In the second section, the authors should provide more information about the dataset.

Thank you for your important comment. We have added more information about obtaining the dataset by adding and rephrasing the following:

“Data examined in this study were collected from all Croatian hospitals in which breast cancer patients are diagnosed and treated. The data were acquired by searching through the hospital information systems during a five-year period, from 01/2017 to 01/2022. Pathohistological and demographic data were obtained for all patients that contained MKB code 50 (code for breast cancer). Pathohistological data was in a standardized format that follows ASCO/CAP guidelines, which all Croatian hospitals use [24].” (lines 122-127)

  1. Hammond, M.E. ASCO-CAP guidelines for breast predictive factor testing: an update. Appl Immunohistochem Mol Morphol 2011, 19, 499-500.

3) The related work section should be improved by including the most recent papers related to breast cancer detection and classification.

 Thank you for your valued comment.

We have added next reference: “50. Sannasi Chakravarthy, S.R.; Rajaguru, H. Automatic Detection and Classification of Mammograms Using Improved Extreme Learning Machine with Deep Learning. IRBM 2022, 43, 49-61.” in the discussion (line 421).

Also, we have added this sentence into introduction: “The latest machine learning studies that focus on breast cancer achieved excellent performances, and are using deep learning techniques with radiomics to classify breast cancer in radiological images or histopathological slides [20-23].” in the introduction. (lines 102-105)

  1. Altameem, A.; Mahanty, C.; Poonia, R.C.; Saudagar, A.K.J.; Kumar, R. Breast Cancer Detection in Mammography Images Using Deep Convolutional Neural Networks and Fuzzy Ensemble Modeling Techniques. Diagnostics 2022, 12.
  2. Muduli, D.; Dash, R.; Majhi, B. Automated diagnosis of breast cancer using multi-modal datasets: A deep convolution neural network based approach. Biomedical Signal Processing and Control 2022, 71, 102825, doi:https://doi.org/10.1016/j.bspc.2021.102825.
  3. Wakili, M.A.; Shehu, H.A.; Sharif, M.H.; Sharif, M.H.U.; Umar, A.; Kusetogullari, H.; Ince, I.F.; Uyaver, S. Classification of Breast Cancer Histopathological Images Using DenseNet and Transfer Learning. Comput Intell Neurosci 2022, 10.

23.Heenaye-Mamode Khan, M.; Boodoo-Jahangeer, N.; Dullull, W.; Nathire, S.; Gao, X.; Sinha, G.R.; Nagwanshi, K.K. Multi- class classification of breast cancer abnormalities using Deep Convolutional Neural Network (CNN). PLoS One 2021, 16.

4) The authors should enhance the quality of the images in the results section. Also include the equation for AUC.

Thank you for your excellent comment.

We have increased the quality of the images from 300 to 500 dpi. Furthermore, we included the AUC formula in the Methods section (lines 207-208): , where and

5) The authors included parameters description for machine learning methods, but I couldn't find a description of the parameters. The authors should include this information in the paper.

Thank you for your valued comment.

We have added the descriptions of optimized hyperparameters in the Methods section (lines 183-193).

“For the random forest we optimized the following hyperparameters: 1) maximal tree depth, 2) minimal number of samples required to split an internal node, 3) minimal number of samples required at a leaf node, 4) number of estimators. Whereas for XGBoost, the following hyperparameters were optimized: 1) maximal tree depth, 2) learning rate, 3) number of estimators, 4) minimum weight required to create a new node (“min_child_weight”), 5) gamma (pseudo-regularization parameter), 6) alpha (L1-regularization of leaf weights), 7) subsample (randomly selected training data prior to fitting to base learner), 8) subsample ratio of columns when constructing each tree (“colsample_bytree”), 9) subsample ratio of columns for each tree depth level (“colsample_bylevel”). Lastly, for logistic regression, we optimized for 1) solver (algorithm to use in the optimization problem), 2) regularization, and 3) regularization strength (C) ”

Reviewer 3 Report

The authors trained and tested several machine learning models on a large cohort of Croatian cancer patient data to assess models' performances in predicting lymph node metastasis. The study aimed to use machine learning to address the misclassification of lymph node metastasis in traditional classification approaches in clinical practice. Despite demonstrating the accuracy of the models, the authors interpreted the trained models using the SHAP approach. The study was well-designed and conducted, and the results and conclusions were appreciable. However, the paper suffers from some issues listed in the comments to the authors.

1.     As also discussed by the authors in lines 396-398, one limitation of the study is the exclusion of a large proportion of samples due to missing values. However, XGBoost could handle missing values. The authors should evaluate XGBoost's performance for the samples that only miss a few features.

2.     As the models were trained on ten features and some significantly differ in the case and control group, e.g., Tumor Size, it is reasonable to see if the trained models outperform predictors that use the most significant feature, Tumor Size. Such univariate-based predictors could serve as baselines for evaluation purposes.

3.     The authors reported the AUROC score as a metric for evaluation, together with some other scores. However, AUROC is only suitable for evaluating balanced data. For imbalanced datasets, Area Under Precision-Recall Curve (AUPRC) or F1 scores should also be reported, especially for the model trained and assessed for the total dataset, whose positive ratio is only 34%.

4.     Line 135: "By applying the NST criteria stated above." However, the only place that mentions the criteria is line 65: "The NST is assigned based on the tumor biology characteristics and radiological findings," which does not provide sufficient information to define NST criteria. Please clarify or correct me if I missed the info.

5.     It's unclear what hyper-parameters were tuned for each of the three models. Specifically, which parameters were tuned for each of the models.

6.     The word "developed" was misused in the paper. The authors developed neither the models nor the SHAP values since these machine learning methods were well developed and maintained by others, e.g., the scikit-learn community. I suppose the authors mean they 'applied' or 'trained.' Please be precise.

7.     Line 63: "positivity of axillary lymph nodes." Although I think the authors are saying "positivity of lymph node metastasis," clarification is needed to avoid confusion.

8.     Line 147: "Random forest and 146 XGBoost were selected because of their high-ranking performances on tabular data" needs a reference.

9.     Line 148: "whereas logistic regression was chosen as a simpler model for comparison" either needs a reference or clarification. It is imprecise to claim that logistic regression is simpler than random forest unless the authors provide a reference or clearly define why it is simpler. Is it because it has fewer parameters or takes less time to train?

10. Line 155: The authors should elaborate on how they performed bootstrap and obtained the statistics.

11. Since line 167, reference number 25, the bracket for reference numbers was changed to a parenthesis. Please be consistent throughout the paper.

12. "Immunofenotype" and "Immunophenotype" were used in the article. The former should be a typo.

Author Response

The authors trained and tested several machine learning models on a large cohort of Croatian cancer patient data to assess models' performances in predicting lymph node metastasis. The study aimed to use machine learning to address the misclassification of lymph node metastasis in traditional classification approaches in clinical practice. Despite demonstrating the accuracy of the models, the authors interpreted the trained models using the SHAP approach. The study was well-designed and conducted, and the results and conclusions were appreciable. However, the paper suffers from some issues listed in the comments to the authors.

1) As also discussed by the authors in lines 396-398, one limitation of the study is the exclusion of a large proportion of samples due to missing values. However, XGBoost could handle missing values. The authors should evaluate XGBoost's performance for the samples that only miss a few features.

Thank you for your valued comment. After excluding the patients that had missing target variable (lymph node metastasis 0/1) we are left with 10540 patients that contain rows with full values and rows with a feature/few features missing. After training and optimizing XGBoost on this dataset, somewhat worse performances were obtained, with an AUC of 0.72 (95%CI 0.64 – 0.76).

Finally, we added the following to the methods section:

(lines 194-196)

“Additionally, since XGBoost can algorithmically deal with missing values, a subanalysis was performed on a dataset with missing values (after dropping the rows that miss the target variable; total N= 10540, NST-criteria N= 1389).”

Also, we added the following to the results:

(lines 275-277)

“When evaluating XGBoost on NST-criteria dataset that contained missing values (n= 1389), somewhat worse performances were obtained, with an AUC of 0.724 (95% CI: 0.654-0.785).” (lines 304-306)

“When evaluating XGBoost on a total dataset that contained missing values (n= 10540), somewhat worse performances were obtained, with an AUC of 0.731 (95% CI: 0.634-0.771).”

2) As the models were trained on ten features and some significantly differ in the case and control group, e.g., Tumor Size, it is reasonable to see if the trained models outperform predictors that use the most significant feature, Tumor Size. Such univariate-based predictors could serve as baselines for evaluation purposes.

Thank you for your comment. The following was added to Methods and Results:

“Furthermore, for evaluation purposes, univariate logistic regression was trained only on one feature (tumour size) as a baseline.” (lines 167- 169)

“Finally, Logistic Regression, has achieved an AUC of 0.763 (95 % CI: 0.683-0.838), while univariate Logistic Regression (trained on “Tumour size”) achieved an AUC of 0.688 (95% CI: 0.626-0.745) (Table 3.).” (lines 259-261)

“Just like with Random Forest and XGBoost, Logistic Regression and Univariate Logistic Regression also scored lower than in the NST criteria group, with a mean AUC of 0.741 (95% CI: 0.706-0.775) and 0.589 (95% CI 0.577-0.614), respectively (Table 4.).” (lines 277- 279)

Predictions were also added to Tables 3. and 4.

Also, the following was added to discussion: “Lastly, both models achieved improved performances when compared to the baseline univariate model that only used tumour size as a predictor.” (lines 370-372)

3) The authors reported the AUROC score as a metric for evaluation, together with some other scores. However, AUROC is only suitable for evaluating balanced data. For imbalanced datasets, Area Under Precision-Recall Curve (AUPRC) or F1 scores should also be reported, especially for the model trained and assessed for the total dataset, whose positive ratio is only 34%.

Thank you for your valued comment. We think that, with 34% of positive cases, the dataset is not highly imbalanced. Also, ranking the classifiers only using the F1-score ignores the negative cases. In clinical terms, that would mean predicting a positive lymph node metastasis status where there is none. Consequently, a patient might receive unnecessary therapy. This is why we opted for using the AUC metric, but in combination with threshold shifting to account for the class imbalance. Maximizing the AUC and applying threshold shifting afterwards has proven very effective in other studies with much lower proportions of positive cases (7.5 % and less than 0.5 %)  (Song, B., Zhang, G., Zhu, W., & Liang, Z. (2014). ROC operating point selection for classification of imbalanced data with application to computer-aided polyp detection in CT colonography. International journal of computer assisted radiology and surgery9(1), 79-89.; Zou, Q., Xie, S., Lin, Z., Wu, M., & Ju, Y. (2016). Finding the best classification threshold in imbalanced classification. Big Data Research5, 2-8.)

As you recommended, aside from the reported metrics, we have also added the F1 score for the highest ranking models. We have added the following at lines 268-271: “At the baseline decision threshold of 0.5, F1-score was 0.750 (95% CI: 0.690-0.812), sensitivity was 0.809 (95% CI 0.718-0.885), specificity 0.570 (95% CI 0.446-0.692), negative predictive value 0.714 (95% CI 0.615-0.820) and the precision (positive predictive value) 0.694 (95% CI: 0.630-0.759).”

We have added the following at lines 295-299: “Concerning XGBoost’s performance on other metrics at the baseline threshold, it achieved F1-score of 0.448 (95% CI: 0.389-0.507), sensitivity of 0.344 (95% CI: 0.289-0.403) and specificity of 0.903 (95%: 0.877-0.926), positive predictive value of 0.607 (95% CI: 0.539-0.680), negative predictive value of 0.761 (95% CI: 0.744-0.778) (Figure 3.).”

Together with lines 300-304:” Lowering the threshold to 0.28 increased the F1-score to 0.581 (95% CI: 0.545-0.618), sensitivity to 0.732 (95% CI: 0.676-0.787), and negative predictive value to 0.854 (95% CI: 0.827-0.881), while specificity decreased to 0.676 (95% CI: 0.637-0.714), and positive predictive value to 0.495 (95% CI: 0.461-0.531).”

4) Line 135: "By applying the NST criteria stated above." However, the only place that mentions the criteria is line 65: "The NST is assigned based on the tumor biology characteristics and radiological findings," which does not provide sufficient information to define NST criteria. Please clarify or correct me if I missed the info.

Thank you for your excellent comment. Mistakenly a paragraph was deleted from the methods when preparing the submit. We have returned the following (lines 150-154):

“Since the model’s target population are patients who would potentially receive neo-adjuvant therapy, we identified those patients from our study population (all patients that initially received surgical treatment) using the following criteria: 1) all tumours with size > 5 cm (irrespective to subtype), 2) tumours with size ≥2cm of triple-negative or HER-2 positive subtype, 3) tumours of inflammatory subtype [29].”

  1. Korde, L.A.; Somerfield, M.R.; Carey, L.A.; Crews, J.R.; Denduluri, N.; Hwang, E.S.; Khan, S.A.; Loibl, S.; Morris, E.A.; Perez, A.; et al. Neoadjuvant Chemotherapy, Endocrine Therapy, and Targeted Therapy for Breast Cancer: ASCO Guideline. J Clin Oncol 2021, 39, 1485-1505, doi:10.1200/jco.20.03399.

5) It's unclear what hyper-parameters were tuned for each of the three models. Specifically, which parameters were tuned for each of the models.

Thank you for your important comment. We have added the following section to Methods (lines 179-188):

“For the random forest we optimized the following hyperparameters: 1) maximal tree depth, 2) minimal number of samples required to split an internal node, 3) minimal number of samples required at a leaf node, 4) number of estimators. Whereas for XGBoost, the following hyperparameters were optimized: 1) maximal tree depth, 2) learning rate, 3) number of estimators, 4) minimum weight required to create a new node (“min_child_weight”), 5) gamma (pseudo-regularization parameter), 6) alpha (L1-regularization of leaf weights), 7) subsample (randomly selected training data prior to fitting to base learner), 8) colsample_bytree (the subsample ratio of columns when con-structing each tree), 9) colsample_bylevel (the subsample ratio of columns for each tree depth level). Lastly, for logistic regression, we optimized for 1) solver (algorithm to use in the optimization problem), 2) regularization, and 3) regularization strength (C). ”

6) The word "developed" was misused in the paper. The authors developed neither the models nor the SHAP values since these machine learning methods were well developed and maintained by others, e.g., the scikit-learn community. I suppose the authors mean they 'applied' or 'trained.' Please be precise.

Thank you for your suggestion, explanation. We have changed “developed” to applied/trained where applicable.

7) Line 63: "positivity of axillary lymph nodes." Although I think the authors are saying "positivity of lymph node metastasis," clarification is needed to avoid confusion.

Thank you for your valued comment. The line was changed to “positivity of metastatic axillary lymph nodes”.

8) Line 147: "Random forest and 146 XGBoost were selected because of their high-ranking performances on tabular data" needs a reference.

Thank you for your important suggestion. The following references were added:

  1. Grinsztajn, L.; Oyallon, E.; Varoquaux, G. Why do tree-based models still outperform deep learning on tabular data? arXiv preprint arXiv:2207.08815 2022
  2. Shwartz-Ziv, R.; Armon, A. Tabular data: Deep learning is not all you need. Information Fusion 2022, 81, 84-90, doi:https://doi.org/10.1016/j.inffus.2021.11.011.

9) Line 148: "whereas logistic regression was chosen as a simpler model for comparison" either needs a reference or clarification. It is imprecise to claim that logistic regression is simpler than random forest unless the authors provide a reference or clearly define why it is simpler. Is it because it has fewer parameters or takes less time to train?

Thank you for your valued comment. We have reformulated the sentence as follows:

“Random forest and XGBoost were selected because of their high-ranking performances on tabular data, whereas logistic regression was chosen as a base classifier for comparison [30]. Furthermore, for evaluation purposes, univariate logistic regression was trained only on one feature (tumour size) as a baseline.”

19) Line 155: The authors should elaborate on how they performed bootstrap and obtained the statistics.

Thank you for your valued comment. The bootstrap was obtained from the test set by performing resampling with replacement.

Hence, the following was added to the previous section about model training and evaluation in the Methods:

“Finally, the performance of the models was assessed on the test set, and the confidence intervals of the performance metrics were estimated using the bootstrap method of resampling with replacement (2000 bootstraped samples).” (lines 190-192)

Additionaly the following was added:

“2000 bootstrap samples obtained by resampling with replacement from the test set were used to determine the mean AUC values and calculate the 95% confidence intervals.” (lines 199-201)

Round 2

Reviewer 1 Report

All my concerns are addressed well.

Reviewer 2 Report

The authors extensively improved the manuscript based on my comments and feedback. The paper is now ready for publication.

Reviewer 3 Report

The authors addressed all my concerns.